# Prevalence of Shigellosis among household contacts of index cases in the EFGH catchment area, Dhaka, Bangladesh

Md Taufiqul Islam☯, Farhana Khanam☯, Md Nazmul Hasan Rajib, Md Ismail Hossen, Syed Qudrat-E-Khuda, Mahzabeen Ireen, Md Golam Firoj, Faisal Ahmmed, Prasanta Kumar Biswas, Amirul Islam Bhuiyan, S. M. Azadul Alam Raz, Md Parvej Mosharraf, Md Taufiqur Rahman Bhuiyan, Firdausi Qadri *

Infectious Diseases Division (IDD), International Centre for Diarrhoeal Disease Research, Bangladesh (icddr,b), Dhaka, Bangladesh

☯ Contributed equally as joint first author

* fqadri@icddrb.org

## Abstract

### Background

This study aimed to estimate the prevalence of shigellosis among household contacts (HHCs) using data from the 'Enterics for Global Health (EFGH)' study, conducted at seven Asian and African countries.

### Methods

In Bangladesh, the EFGH study was conducted in Maniknagar, Dhaka, to determine the burden of shigellosis among children aged 6–35 months. HHCs of *Shigella*-positive patients (index cases) were enrolled in this study. Stool specimens form contacts were collected within 7 days of enrollment of index cases and culture and qPCR were performed. Sociodemographic and behavioral information were obtained to identify risk factors.

### Results

A total of 400 HHCs of 118 index cases were enrolled, of which 36 (9%) were positive for *Shigella* spp. by culture, while qPCR revealed 21% (42/200) of contacts had *Shigella* infections. Individuals who failed to reheat meals before consumption had a two-fold higher risk of shigellosis compared to those who reheated meals, although other sociodemographic and behavioral factors did not show any significant association.

**Data availability statement:** A de-identified analytical data set will be made available upon requests directed to the Institutional Review Board (IRB) coordinator of the icddr,b at info@icddrb.org. Only after approval of a proposal data can be shared through a secure online platform. Approval of the proposal will be subject to scientific review by the institutional review board at icddr,b. Sharing of data will also be subject to the published data access rules of the icddr,b. The requestor will need to sign a standard data access agreement required by the icddr,b.

**Funding:** This study was supported by the Bill & Melinda Gates Foundation [INV-045988]. The funders had no role in study design, data collection and analysis, decision to publish, or preparation of the manuscript. No authors received any salary from the funder.

**Competing interests:** The authors have declared that no competing interests exist.

## Conclusions

The study revealed a high burden of asymptomatic *Shigella*-infected individuals, emphasizing the need for systematic surveillance to assess the burden and develop effective preventive strategies to prevent the spread of the disease.

## Author summary

Asymptomatic individuals can play a key role in transmitting *Shigella* within communities in endemic settings. We assessed the prevalence and risk factors of *Shigella* infection among household contacts of culture-confirmed index cases in an urban slum of Dhaka, Bangladesh. Our study revealed a high proportion of household contacts were infected, with 97% of them excreting the organism asymptomatically. Index cases were children aged 6–35 months, and household members represented a range of age groups, with 75% being adults. Failure to reheat meals before consumption and the use of non-flush toilets were significant risk factors. These findings highlight the urgent need for interventions that extend beyond symptomatic case management, emphasizing safe food handling practices and improved sanitation to interrupt transmission chains effectively.

## Background

Shigellosis is a severe intestinal infection caused by the gram-negative bacterium *Shigella* spp., primarily transmitted through the fecal-oral route [1]. Shigellosis is one of the leading causes of diarrheal illness worldwide, contributing to substantial morbidity and mortality, particularly among children under five years of age [2]. This disease represents a major public health challenge, particularly in low- and middle-income countries like Bangladesh, where inadequate sanitation and limited access to clean water exacerbate its spread. In a hospital-based study, the incidence of shigellosis among children under five in a densely populated urban area of Dhaka, Bangladesh was 4.6 cases per 100 person-years [3,4]. A previous study in rural Bangladesh highlighted the increased risk of *Shigella* transmission among infected individuals' household contacts, underscoring this disease's widespread nature [2]. Another retrospective cohort study conducted in Amsterdam between 2002 and 2009 showed that the laboratory-confirmed *Shigella* infection in high-risk household contacts of primary cases was 7.4% [5].

The Enterics for Global Health (EFGH) *Shigella* burden study, conducted in urban Dhaka from 2022 to 2024, aimed to estimate the incidence rates of *Shigella*-attributed diarrhea in children aged 6–35 months, documenting health outcomes and informing public health strategies and interventions. A sub-study was planned using the EFGH platform. The objective of the study was to estimate the proportion of *Shigella* infection among household contacts and also to identify the risk factors associated with the infection among household contacts (HHCs).

## Methods

### Ethical clearance

This study (PR 23034) was approved by the Research Review Committee (RRC) and Ethical Review Committee (ERC) of International Centre for Diarrhoeal Disease Research, Bangladesh, icddr,b. We obtained informed written consent or assent from all eligible study participants or their legal guardian prior to the enrollment.

### Enrollment of study participants

The EFGH study was conducted in wards 6, 7, 48, 49, 50, 71, and 72 of Dhaka South City Corporation, Bangladesh, from the EFGH *Shigella* surveillance sites. Patients aged 6–35 months presented with diarrhea to the icddr,b Dhaka hospital, the Mugda Medical College Hospital, and the EFGH field office in the Maniknagar area were enrolled in the EFGH study. A rectal swab was collected from them for microbiological culturing. A culture-confirmed *Shigella* patient was considered as an index case and their household contacts were selected for this sub-study. The HHCs were individuals of all ages who permanently resided at the time of the study in the same household as the index cases, shared food from the same kitchen as the index cases, and were present for at least 7 of the last 14 days.

All HHCs of *Shigella*-positive index cases were approached and enrolled in this study from April 2023 to June 2024 after completing the proper consenting procedure. Enrollment occurred within 7 days of receiving the culture reports of EFGH index cases. Study staff visited the households on the 6th day and provided stool containers to the parents or guardians of index cases to collect stool specimens from all household members on day 7. The participants were reminded by phone calls or household visits to ensure the collection of day 7 stool specimens. In addition, if any household members developed acute watery diarrhea (≥3 abnormally loose or watery stools in 24 hours) within seven days, a stool specimen was obtained from them. No further stool collection was carried out.

### Specimen collection, preparation and transportation

Stool specimens were collected in stool containers from the contacts of the index cases. A stool swab was prepared from a stool specimen collected from HHCs and placed dry in a transport tube (2 ml screw top vial compatible with a bead beater) for TaqMan Array Card (TAC) testing. All specimens were then immediately transported to the icddr,b laboratory, maintaining a temperature of 2–8˚C [6].

### Laboratory methods

Stool specimens were streaked directly onto MacConkey agar and Xylose Lysine Deoxycholate agar media and incubated at 37°C for 18–24 hours. Morphologically suspicious *Shigella* colonies were inoculated onto Triple Sugar Iron (TSI), Motility Indole Urea (MIU) and Tryptic Soy Agar (TSA). *Shigella* isolates were then confirmed by serotyping test using commercially available antisera (Denka Seiken Co., Ltd., Tokyo, Japan) for agglutination. The antibiotic susceptibility testing (AST) of confirmed *Shigella* isolates was done by the Kirby-Bauer disk diffusion method and zone diameters were recorded and classified as susceptible, resistant, or intermediate based on the Clinical and Laboratory Standards Institute guideline [7]. AST was performed for various antibiotics, including ampicillin, azithromycin, ceftriaxone, ciprofloxacin, nalidixic acid, pivmecillinam, and trimethoprim/sulfamethoxazole.

The QIAamp Fast DNA Stool Mini kit (Qiagen, Valencia, CA) was used to extract nucleic acid from stool specimens for TAC testing. The quantitative analysis was performed to determine the likely attribution of *Shigella* and other particular etiologies of diarrhea using the QuantStudio 7 Flex. A customized TaqMan Array Card (TAC) was used, which had a total of 82 targets. Among them, 37 targets were for bacteria, 8 for viruses, 1 for fungi, 5 for protozoa, and 4 targets for extraction and plate control. After completion of the plate run, the dataset was uploaded to the Multi-Schema Information Capture database platform for analysis. *Shigella* was targeted using ipaH/*Shigella* EIEC, and the cut-off for positivity was set at a

quantification cycle (Cq) ≤29.8. When the Cq value exceeded 29.8, *Shigella* serotypes were not pursued further. For other pathogens, the Cq cut-offs were the same as that was used in previous studies were applied [8].

### Data capturing and management

Comprehensive documentation of each participant was maintained throughout the study. Data on demographic, socioeconomic, clinical characteristics, and water sanitation and hygiene (WASH) behaviour and practices were collected using structured case report forms and entered into the electronic data capturing system and then transferred to the SQL-Server of icddr,b. Errors in the database were detected and corrected following a standard operating procedure of maintaining an audit trail.

### Statistical analysis

The prevalence of *Shigella* infection among household contacts was determined using numbers and percentages and the prevalence of households with *Shigella* infection was calculated based on various household-level demographic, socio-economic, and WASH characteristics. The numerator was the number of households with *Shigella* infection, and the denominator was the total number of households at risk. A 95% confidence interval (CI) for the prevalence was calculated using binomial proportions of the Wilson method. Bar diagrams and pie charts were used to present the summarized TAC results for various organisms, serotypes of *Shigella* and the antimicrobial susceptibility patterns. Association between *Shigella* infection among HHCs and individual- and household-level demographic, socio-economic, and WASH characteristics were assessed using the Chi-square and the Fisher's exact test.

Logistic regression was carried out to measure the relationship between predisposing factors in household contacts and *Shigella* infection. Exponential coefficients of the binomial Logistic regression models with log link function have been used and presented as risk ratios (RR) with a corresponding 95% CI. Variables with a significance level of ≤0.2 in univariate analysis were included in the multiple regression model as independent variables. All tests were two-tailed, where p-value <0.05 was considered statistically significant. The statistical analysis was conducted using R (version 4.3.3). For plots and figures, "ggplot2" package and Microsoft Excel were used.

## Results

### Assembly of the participants

We identified 118 culture-confirmed *Shigella* index cases in the EFGH study. A total of 443 household members of the EFGH index cases were approached for enrollment in this study. Of these, 24 refused to participate, and 19 were not enrolled in this study as they developed acute watery diarrhea (AWD) within one day of the index case's enrollment (Fig 1). Among 400 household members enrolled in the study, 205 (51%) were female, and 299 (75%) were adults.

### *Shigella* infection among household contacts by using microbiological culture of stool specimens

Microbiological culture of stool samples collected from 400 enrolled household contacts revealed that 36 individuals (9%) were *Shigella*-positive. Of these, only one had diarrhea before enrollment, while the remaining 35 cases were asymptomatic (Fig 1). The *Shigella* isolates consisted of four species: *Shigella flexneri* (n = 15), *Shigella sonnei* (n = 10), *Shigella boydii* (n = 7), and *Shigella dysenteriae* (n = 4). Among the *S. flexneri* isolates, subtyping revealed a range of serotypes, including *S. flexneri* 2a (n = 4), *S. flexneri* 4a (n = 1), *S. flexneri* 1a (n = 1), and 9 non-typable *S. flexneri* isolates (Table 1). Not all household members with stool cultures positive for *Shigella* had the same serogroup as the index case: 11 shared the same serogroups, while the others had different ones. 29 households out of 118 had an infection. 2 members from 5 households and 3 members from one HH were infected.

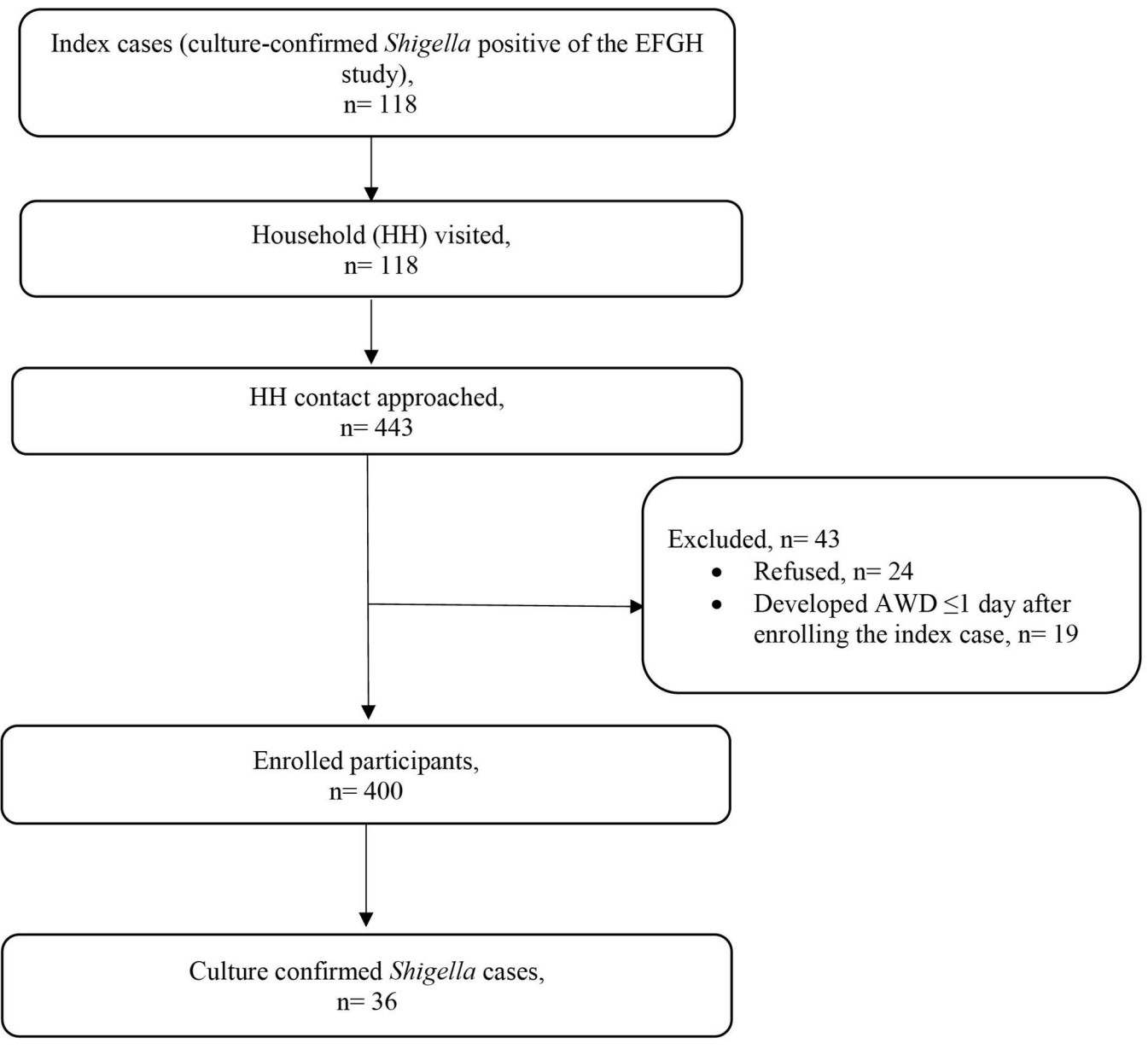

**Fig 1.  CONSORT diagram of household contact study.**

### Risk factors of *Shigella* infection among HH contacts

For individual-level characteristics, there were no significant differences in *Shigella* infection by age group, gender, or religion. Education level showed a trend, though the difference was not statistically significant (p = 0.063): there were more *Shigella*-positive cases among individuals with secondary education (19%) compared to those with higher education (6%). Occupational status, food consumption outside the home, and use of soap for personal hygiene did not significantly differ between *Shigella*-positive cases and *Shigella*-negative individuals. Notably, individuals who did not heat their meals

**Table 1. Proportion of *Shigella* species detected by culture and pathogens detected by TAC among HH contacts of index cases.**

| Culture results | No. of positive cases | Percentage |
|---|---|---|
| ***Shigella flexneri*** | **15** | **3.8** |
| *Shigella flexneri 2a* | 4 | 26.7 |
| *Shigella flexneri 4a* | 1 | 6.7 |
| *Shigella flexneri 1a* | 1 | 6.7 |
| *Shigella flexneri non-typable* | 9 | 60 |
| ***Shigella sonnei*** | **10** | **2.5** |
| ***Shigella boydii*** | **7** | **1.8** |
| ***Shigella dysenteriae*** | **4** | **1** |
| **TaqMan Array Card (TAC) results** | **No. of positive cases** | **Percentage** |
| EAEC | 139 | 69.5 |
| EPEC | 125 | 62.5 |
| ETEC_LT | 100 | 50 |
| *Giardia* | 96 | 48 |
| ETEC_ST | 93 | 46.5 |
| *Shigella*_EIEC | 78 | 39 |
| *C infans* | 51 | 25.5 |
| *C jejuni coli* | 23 | 11.5 |
| Adenovirus | 17 | 8.5 |
| Plesiomons | 16 | 8 |
| *E.bieneusi* | 14 | 7 |
| Sapovirus 124 | 12 | 6 |
| STEC | 9 | 4.5 |
| *Cryptosporidium* | 8 | 4 |
| Aeromonas | 7 | 3.5 |
| *Vibrio cholerae* | 6 | 3 |
| *C_Upsalensis* | 6 | 3 |
| *H pylori* | 4 | 2 |
| Norovirus G-II | 4 | 2 |
| Rotavirus | 3 | 1.5 |
| Norovirus G-I | 2 | 1 |
| *Salmonella* | 1 | 0.5 |
| SARS-CoV-2 | 1 | 0.5 |
| Astrovirus | 1 | 0.5 |
| *Cyclospora* | 1 | 0.5 |

before consumption had a significantly higher proportion of *Shigella* positivity (19%) compared to those who did (8%) (p = 0.041).

At the household level, no significant association between family size and *Shigella* positivity was observed. Duration of residence in the current household (more than one year or less than one year) did not show significant differences between *Shigella*-positive and *Shigella*-negative participants. Additionally, construction materials of the roof, walls, and floors had no significant association with *Shigella* positivity. Sanitation facilities, including toilet type, toilet sharing, kitchen sharing, and water sources, also showed no significant differences. Water adequacy and treatment practices, along with waste disposal methods, did not reveal notable associations with *Shigella* infection among household members (Table 2).

**Table 2.** Household and individual level demographic, socio-economic, and WASH characteristics of the *Shigella* positive and negative HH contacts.

| Household level | | | | |
|---|---|---|---|---|
| **Factors** | Labels | *Shigella* positive n=36 | *Shigella* negative n=364 | p-value |
| **Number of family member** | <4 | 20 (12%) | 153 (88%) | 0.158 |
| | ≥4 | 16 (7%) | 211 (93%) | |
| **Number of persons live per room** | ≤2 | 14 (9%) | 149 (91%) | 0.861 |
| | >2 | 22 (9%) | 215 (91%) | |
| **Living in this household till** | ≤1 year | 18 (10%) | 159 (90%) | 0.486 |
| | >1 year | 18 (8%) | 205 (92%) | |
| **Construction materials of the roof** | Brick/Cement | 22 (8%) | 249 (92%) | 0.455 |
| | Others | 14 (11%) | 115 (89%) | |
| **Construction materials of the wall** | Brick/Cement | 31 (9%) | 323 (91%) | 0.588 |
| | Others | 5 (11%) | 41 (89%) | |
| **Construction materials of the floor** | Brick/Cement | 36 (9%) | 362 (91%) | >0.999 |
| | Others | 0 (0%) | 2 (100%) | |
| **Toilet type** | Flush or pour flush | 35 (9%) | 363 (91%) | 0.172 |
| | Non-flush | 1 (50%) | 1 (50%) | |
| **Shared toilet** | Yes | 17 (9%) | 166 (91%) | 0.863 |
| | No | 19 (9%) | 198 (91%) | |
| **Shared kitchen** | Yes | 18 (8%) | 195 (92%) | 0.728 |
| | No | 18 (10%) | 169 (90%) | |
| **Source of water** | Tube-well/Bottled/ ATM booth | 9 (8%) | 111 (92%) | 0.571 |
| | Supply/Tanker/Public tap/Rain/River etc. | 27 (10%) | 253 (90%) | |
| **Amount of water adequate** | Yes | 36 (9%) | 364 (91%) | – |
| | No | – | – | |
| **Treatment of water** | Boiled/filtered/ chemical | 18 (9%) | 190 (91%) | 0.862 |
| | Not treated | 18 (9%) | 174 (91%) | |
| **Water available 24 hours and 7 days** | Yes | 36 (9%) | 364 (91%) | – |
| | No | – | – | |
| **Distance of water source from toilet/sewerage system in feet** | Within 3 feet | 19 (8%) | 231 (92%) | 0.212 |
| | More than 3 feet | 17 (11%) | 133 (89%) | |
| **Place of waste disposal** | Fixed | 36 (9%) | 359 (91%) | >0.999 |
| | Indiscriminate | 0 (0%) | 5 (100%) | |
| Individual level | | | | |
| **Age** | <5 years | 2 (9%) | 21 (91%) | 0.856 |
| | 5 to 17 years | 8 (10%) | 70 (90%) | |
| | ≥18 years | 26 (9%) | 273 (91%) | |
| **Gender** | Male | 17 (9%) | 178 (91%) | 0.863 |
| | Female | 19 (9%) | 186 (91%) | |
| **Religion** | Muslim | 36 (9%) | 364 (91%) | – |
| **Educational status** | Higher educated | 4 (6%) | 62 (94%) | 0.063 |
| | Secondary* | 5 (19%) | 21 (81%) | |
| | Primary | 7 (5%) | 124 (95%) | |
| | Child/Illiterate | 20 (11%) | 157 (89%) | |

*(Continued)*

**Table 2.** (Continued)

| Household level | | | | |
|---|---|---|---|---|
| **Factors** | Labels | *Shigella* positive n = 36 | *Shigella* negative n = 364 | p-value |
| **Occupational status** | Work Outside | 14 (8%) | 161 (92%) | 0.834 |
| | House maker | 12 (10%) | 111 (90%) | |
| | Unemployed/ Child/Student | 10 (10%) | 92 (90%) | |
| **Taken food outside home in last 7 days** | Yes | 20 (9%) | 207 (91%) | >0.999 |
| | No | 16 (9%) | 157 (91%) | |
| **Heat meal before intake** | Yes | 28 (8%) | 329 (92%) | 0.041 |
| | No | 8 (19%) | 35 (81%) | |
| **Use soap after defecation** | Yes | 35 (9%) | 356 (91%) | 0.576 |
| | No | 1 (11%) | 8 (89%) | |
| **Use soap before taking food** | Yes | 36 (9%) | 359 (91%) | >0.999 |
| | No | 0 (0%) | 5 (100%) | |

*Secondary education includes schooling for 5 years, classes from VI to X.

The multivariate analysis revealed that secondary education posed a higher risk for *Shigella* infection among household members, with an adjusted relative risk (RR) of 3.19 (95% CI: 0.95, 10.68; p = 0.060), indicating a threefold higher risk compared to higher educational levels. Additionally, failing to reheat meals before consumption showed a significant risk (RR = 2.27, 95% CI: 1.07, 4.83; p = 0.034) for being infected. Also, the type of toilet used indicated that non-flush users had a higher risk for *Shigella* infection, with an adjusted relative risk of 6.16 (95% CI: 1.31, 29.03; p = 0.021). Additionally, the distance of >3 feet between the water source and toilet/sewerage system was associated with a risk of infection (RR = 2.35, 95% CI: 0.70, 2.59), but this association was not statistically significant. However, the analysis did not show any significant effect of using soap before eating in reducing the risk of infection (Table 3).

### Antimicrobial resistance pattern of isolated *Shigella*

Antibiotic susceptibility profiles revealed distinct resistance patterns among the four *Shigella* species. All isolates were uniformly susceptible to pivmecillinam. *S. flexneri* exhibited high resistance to ampicillin (60.0%) and notable resistance to nalidixic acid (46.7%). *S. sonnei* showed complete resistance to ciprofloxacin and nalidixic acid (100.0%), and high resistance to azithromycin (70.0%). *S. boydii* was highly susceptible to most antibiotics but showed resistance to nalidixic acid (85.7%). *S. dysenteriae* was susceptible to all antibiotics except nalidixic acid, in which 50.0% of isolates were resistant. (Fig 2).

### Pathogens identified from TaqMan Array Card (TAC) assay

Among the 400 participants, the first 200 stool specimens collected were tested using the TAC assay. *Shigella* was identified in 42 (21%) of the samples, with the majority (69%, 29 samples) classified as *Shigella flexneri*, while 13 (31%) samples were identified as *Shigella sonnei*. Among the *Shigella flexneri* isolates, the most prevalent serotypes were *Shigella flexneri* 1a, 2a, 2b, 4a, 5a, 6, 7a and X in addition to *Shigella*, the analysis detected 26 other organisms, including enterotoxigenic *Escherichia coli,* adenovirus, rotavirus and *cryptosporidium* spp. A total of 5 (38.4%) samples had co-infection with *Shigella* (Table 1).

**Table 3. Predisposing factors in household contacts for developing *Shigella* infection.**

| Factors | Labels | Crude RR (95%CI) | p value | Adj. RR (95%CI) | p value |
|---|---|---|---|---|---|
| **Age** | ≥18 years | Ref. | – | – | – |
| | 5 to 17 years | 1.18 (0.56, 2.5) | 0.667 | – | – |
| | <5 years | 1 (0.25, 3.95) | >0.999 | – | – |
| **Gender** | Male | Ref. | – | – | – |
| | Female | 1.06 (0.57, 1.98) | 0.848 | – | – |
| **Educational status** | Higher educated | Ref. | – | Ref. | – |
| | Secondary | 3.17 (0.92, 10.9) | 0.067 | 3.19 (0.95, 10.68) | 0.060 |
| | Primary | 0.88 (0.27, 2.9) | 0.836 | 0.69 (0.21, 2.28) | 0.542 |
| | Child/Illiterate | 1.86 (0.66, 5.25) | 0.238 | 1.32 (0.46, 3.78) | 0.608 |
| **Occupational status** | Work Outside | Ref. | – | – | – |
| | House maker | 1.22 (0.58, 2.55) | 0.597 | – | – |
| | Unemployed/ Child/Student | 1.23 (0.57, 2.66) | 0.607 | – | – |
| **Taken food outside home in last 7 days** | No | Ref. | – | – | – |
| | Yes | 0.95 (0.51, 1.78) | 0.879 | – | – |
| **Heat meal before intake** | Yes | Ref. | – | Ref. | – |
| | No | 2.37 (1.16, 4.87) | 0.019 | 2.27 (1.07, 4.83) | 0.034 |
| **Use soap after defecation** | Yes | Ref. | – | – | – |
| | No | 1.24 (0.19, 8.09) | 0.821 | – | – |
| **Use soap before taking food** | Yes | Ref. | – | – | – |
| | No | 0 (0, Inf) | 0.989 | – | – |
| **Number of family member** | <4 | Ref. | – | Ref. | – |
| | ≥4 | 0.61 (0.33, 1.14) | 0.122 | 0.62 (0.33, 1.17) | 0.142 |
| **Number of persons live per room** | ≤2 | Ref. | – | – | – |
| | >2 | 1.08 (0.57, 2.05) | 0.812 | – | – |
| **Living in this household till** | ≤1 year | Ref. | – | – | – |
| | >1 year | 0.79 (0.43, 1.48) | 0.467 | – | – |
| **Construction materials of the roof** | Brick/Cement | Ref. | – | – | – |
| | Others | 1.34 (0.71, 2.53) | 0.371 | – | – |
| **Construction materials of the wall** | Brick/Cement | Ref. | – | – | – |
| | Others | 1.24 (0.51, 3.03) | 0.635 | – | – |
| **Construction materials of the floor** | Brick/Cement | Ref. | – | – | – |
| | Others | 0 (0, Inf) | 0.989 | – | – |
| **Toilet type** | Flush or pour flush | Ref. | – | Ref. | – |
| | Non-flush | 5.69 (1.37, 23.56) | 0.017 | 6.16 (1.31, 29.03) | 0.021 |
| **Shared toilet** | No | Ref. | – | – | – |
| | Yes | 1.06 (0.57, 1.98) | 0.853 | – | – |
| **Shared kitchen** | No | Ref. | – | – | – |
| | Yes | 0.88 (0.47, 1.64) | 0.682 | – | – |
| **Source of water** | Tube-well/Bottled/ ATM booth | Ref. | – | – | – |
| | Supply/Tanker/Public tap/Rain/ River etc. | 1.29 (0.62, 2.65) | 0.496 | – | – |
| **Treatment of water** | Boiled/filtered/ chemical | Ref. | – | – | – |
| | Not treated | 1.08 (0.58, 2.02) | 0.801 | – | – |
| **Distance of water source from toilet/ sewerage system in feet** | Within 3 feet | Ref. | – | Ref. | – |
| | More than 3 feet | 1.49 (0.8, 2.78) | 0.208 | 2.35 (0.70, 2.59) | 0.369 |
| **Place of waste disposal** | Fixed | Ref. | – | – | – |
| | Indiscriminate | 0 (0, Inf) | 0.989 | – | – |

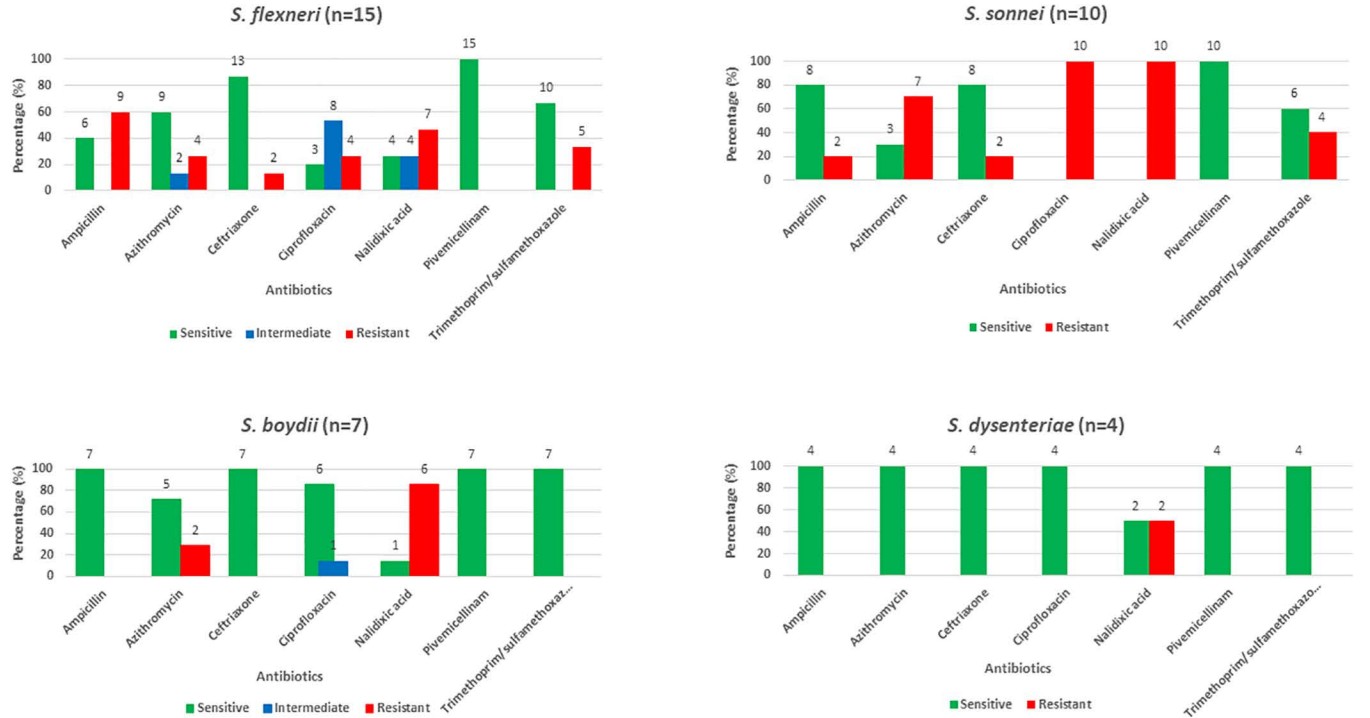

**Fig 2. Antimicrobial susceptibility pattern of isolated *Shigella* strains among the household contacts.**

## Discussion

There is a substantial burden of (9%) *Shigella* infection among asymptomatic household members of culture-confirmed *Shigella* index cases, identified in the EFGH Bangladesh site conducted between June 2022 and October 2024 [7]. The high prevalence of asymptomatic cases underscores the potential for asymptomatic carriage, which may play a crucial role in transmission of *Shigella* within families and communities, contributing to a high burden of *Shigella* infection in this setting. Our findings also indicate that households with identified index cases are at particularly high risk of shigellosis. However, the differences in *Shigella* serogroups observed between some index cases and their household members suggest that these infections were not directly related to the index cases, or that secondary transmission did not occur.

Similarly, studies conducted in Amsterdam from 1992 to 1998 reported 8% rate of *Shigella* infection among household members and asymptomatic individuals [9]. Between April and August 2008, an outbreak of *Shigella sonnei* infection was documented in the Orthodox Jewish community of Antwerp, Belgium, with a household attack rate of 8.5% [10]. A study conducted in Bangladesh on the intra-family transmission of enteric diseases reported that the intra-familial *Shigella* transmission rate was 8% [11]. Furthermore, a study in rural Bangladesh indicated that household members were likely to transmit *Shigella* infection within the week following the initial diagnosis of *Shigella*-infected patients [2].

The antibiotic susceptibility testing (AST) of *Shigella* isolates revealed higher sensitivity to pivmecillinam, ceftriaxone, and trimethoprim/sulfamethoxazole, with a significant proportion of isolates demonstrating resistance to nalidixic acid and ciprofloxacin. These findings will be helpful in selecting appropriate antibiotics for treating *Shigella*-infected patients, particularly in settings where AST facilities are inadequate. A study conducted in Bangladesh showed similar patterns, with 93% sensitivity to pivmicillinam, 96.8% sensitivity to ceftriaxone, and 93% resistance to nalidixic acid. Additionally, a meta-analysis based on studies from Bangladesh revealed resistance patterns against *Shigella*, with 38.8% resistance

to azithromycin, 34.5% resistance to ampicillin, 31.1% resistance to ciprofloxacin, and 10.8% resistance to ceftriaxone, which was similar to our findings [12,13].

Our analysis revealed several significant risk factors for *Shigella* infection among HHCs. Education up to secondary education level only was associated with a higher risk, with an adjusted relative risk (RR) of 3.19, indicating a threefold higher risk compared to individuals with higher educational levels. This finding aligns with a previous study identifying lower educational levels as a risk factor for various infectious diseases [14]. Additionally, failing to reheat meals before consumption was associated with a significant risk (RR = 2.27), consistent with studies highlighting the importance of proper food handling and reheating practices in preventing foodborne illnesses [15]. The type of toilet used also appeared as a risk factor, with non-flush toilets posing a significant risk. This finding is supported by a study showing that inadequate sanitation facilities contribute to the spread of enteric diseases [16].

Our study had several limitations. Firstly, it was conducted in the densely populated urban area of Dhaka city, which may not accurately reflect transmission dynamics in rural areas of Bangladesh or less inhabited regions in LMICs globally. Secondly, the use of self-reported data for certain variables in the risk factor analysis, such as hygiene practices, could introduce reporting bias. Lastly, although the sample size was calculated based on the primary objective (the proportion of infection among household members), this may have limited power to identify additional risk factors. Despite these limitations, our findings have great implications, highlighting a high proportion of asymptomatic *Shigella* infections. This contributes to a sustained high burden of infection, with morbidity and subsequent consequences such as growth failure, hospitalization, and death among young children.

In conclusion, the evidence generated from this study shows the spread of infection within households and may be leading to carriage. This information will support policymakers in the development of comprehensive and targeted preventive strategies, ultimately leading to better health outcomes and a reduced disease burden by effectively interrupting the transmission chains in the community.

## Acknowledgments

The icddr,b is grateful to the governments of Bangladesh and Canada for providing core/unrestricted support. We would like to thank Patricia B. Pavlinac, Kirk Tickell, Stephanie Edlund Cho, Sonia Rao, Hannah Atlas from the University of Washington and Beth Tippett Barr from Nyanja Health Research Institute, Eric Houpt and Queen Saidi from the University of Virginia.

## Author contributions

**Conceptualization:** Md Taufiqul Islam, Farhana Khanam.

**Data curation:** Md Golam Firoj, Faisal Ahmmed, Prasanta Kumar Biswas.

**Formal analysis:** Md Taufiqul Islam, Farhana Khanam, Md Golam Firoj, Faisal Ahmmed.

**Funding acquisition:** Md Taufiqul Islam.

**Investigation:** Md Taufiqul Islam, Farhana Khanam, S.M. Azadul Alam Raz, Md. Parvej Mosharraf.

**Methodology:** Md Taufiqul Islam, Farhana Khanam.

**Project administration:** Md Taufiqul Islam, Farhana Khanam, Md Nazmul Hasan Rajib, Md Ismail Hossen, Syed Qudrat-E-Khuda, Mahzabeen Ireen, Firdausi Qadri.

**Resources:** Firdausi Qadri.

**Software:** Prasanta Kumar Biswas.

**Supervision:** Md Taufiqul Islam, Farhana Khanam, Md Nazmul Hasan Rajib, Md Ismail Hossen, Syed Qudrat-E-Khuda, Mahzabeen Ireen, Firdausi Qadri.

**Validation:** Md Golam Firoj, Faisal Ahmmed.

**Visualization:** Md Taufiqul Islam, Farhana Khanam.

**Writing – original draft:** Md Taufiqul Islam, Farhana Khanam.

**Writing – review & editing:** Md Taufiqul Islam, Farhana Khanam, Md Nazmul Hasan Rajib, Md Ismail Hossen, Syed Qudrat-E-Khuda, Mahzabeen Ireen, Md Golam Firoj, Faisal Ahmmed, Prasanta Kumar Biswas, Amirul Islam Bhuiyan, S.M. Azadul Alam Raz, Md. Parvej Mosharraf, Md Taufiqur Rahman Bhuiyan, Firdausi Qadri.

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
