## [Decision Letter · Decision Letter 0]

29 Jul 2025

Prevalence of Shigellosis among household contacts of index cases in the EFGH catchment area, Dhaka, Bangladesh

Dear Dr. Qadri,

Thank you for submitting your manuscript to PLOS Neglected Tropical Diseases. After careful consideration, we feel that it has merit but does not fully meet PLOS Neglected Tropical Diseases's publication criteria as it currently stands. Therefore, we invite you to submit a revised version of the manuscript that addresses the points raised during the review process.

Please submit your revised manuscript within 60 days Sep 27 2025 11:59PM. If you will need more time than this to complete your revisions, please reply to this message or contact the journal office at plosntds@plos.org. Please include the following items when submitting your revised manuscript:

We look forward to receiving your revised manuscript.

Kind regards,

Nicholas J Mantis

Academic Editor

Stuart Blacksell

Section Editor

Shaden Kamhawi

co-Editor-in-Chief

Paul Brindley

co-Editor-in-Chief

**Additional Editor Comments:**

There was a split decision between the two reviewers about the merit of the manuscript. While both Reviewers recognize the significance of the study, Reviewer 1 raised concerns about the analysis conducted to define relationships between the index cases and the household cases. The reviewer suggested you re-analyze and re-write the paper from the data set, but with more consideration of the types of analysis needed. This comment in particular influenced the decision for a major revision.

**Journal Requirements:**

At this stage, the following Authors/Authors require contributions: Md Taufiqul Islam, Farhana Khanam, Md Nazmul Hasan Rajib, Md Ismail Hossen, Syed Qudrat-E-Khuda, Mahzabeen Ireen, Md Golam Firoj, Faisal Ahmmed, Prasanta Kumar Biswas, Amirul Islam Bhuiyan, S.M. Azadul Alam Raz, Md. Parvej Mosharraf, Md Taufiqur Rahman Bhuiyan, and Firdausi Qadri. Please ensure that the full contributions of each author are acknowledged in the "Add/Edit/Remove Authors" section of our submission form.

3) Tables should not be uploaded as individual files. Please remove these files and include the Tables in your manuscript file as editable, cell-based objects. For more information about how to format tables, see our guidelines:

https://journals.plos.org/plosntds/s/tables 

4) We note that your Data Availability Statement is currently as follows: "All relevant data are within the paper and its supporting information files.". Please confirm at this time whether or not your submission contains all raw data required to replicate the results of your study. Authors must share the “minimal data set” for their submission. PLOS defines the minimal data set to consist of the data required to replicate all study findings reported in the article, as well as related metadata and methods (https://journals.plos.org/plosone/s/data-availability#loc-minimal-data-set-definition).

5) Please provide a detailed Financial Disclosure statement. This is published with the article. It must therefore be completed in full sentences and contain the exact wording you wish to be published.

1) Please clarify all sources of financial support for your study. List the grants, grant numbers, and organizations that funded your study, including funding received from your institution. Please note that suppliers of material support, including research materials, should be recognized in the Acknowledgements section rather than in the Financial Disclosure

2) State the initials, alongside each funding source, of each author to receive each grant. For example: "This work was supported by the National Institutes of Health (####### to AM; ###### to CJ) and the National Science Foundation (###### to AM)."

3) State what role the funders took in the study. If the funders had no role in your study, please state: "The funders had no role in study design, data collection and analysis, decision to publish, or preparation of the manuscript."

4) If any authors received a salary from any of your funders, please state which authors and which funders..

6) Your current Financial Disclosure states, "The author(s) received no specific funding for this work.".

However, your funding information on the submission form indicates  receiving funds.. 

Please indicate by return email the full and correct funding information for your study and confirm the order in which funding contributions should appear. Please be sure to indicate whether the funders played any role in the study design, data collection and analysis, decision to publish, or preparation of the manuscript.

**Reviewers' Comments:**

Reviewer's Responses to Questions

**Key Review Criteria Required for Acceptance?**

**Methods**

-Are the objectives of the study clearly articulated with a clear testable hypothesis stated?

-Is the study design appropriate to address the stated objectives?

-Is the population clearly described and appropriate for the hypothesis being tested?

-Is the sample size sufficient to ensure adequate power to address the hypothesis being tested?

-Were correct statistical analysis used to support conclusions?

-Are there concerns about ethical or regulatory requirements being met?

Reviewer #1: The objectives are clear - that is to carry out a family study of shigellosis. However, there is no testable hypothesis. The population could be better described by age, sex, season.

Reviewer #2: The methods are clearly described and are appropriate for addressing the objectives of this study.

**Results**

-Does the analysis presented match the analysis plan?

-Are the results clearly and completely presented?

-Are the figures (Tables, Images) of sufficient quality for clarity?

Reviewer #1: The analysis presented match the plan, however, the analysis could have gone into more depth and detail.

i felt the figures 2 and 4 are not needed.

Reviewer #2: The results match the analysis plan and are supported by the data presented. The tables and figures are clear and helpfully illustrate the key findings.

**Conclusions**

-Are the conclusions supported by the data presented?

-Are the limitations of analysis clearly described?

-Do the authors discuss how these data can be helpful to advance our understanding of the topic under study?

-Is public health relevance addressed?

Reviewer #1: The authors found many shigella infections in household members of index patients, but it is difficult to know what this means.

Reviewer #2: The conclusions are clearly presented and are well-supported by the data provided.

**Editorial and Data Presentation Modifications?**

Reviewer #1: Editorial comments (the numbers refer to the line numbers).

37, Why mention vaccinations in the abstract when there are no licensed vaccines

83. Is there validation that a cold temp is needed. Could it be even harmful to maintain survival of shigellae?

85 stool specimens (plural) were…

92 reference(s) needed for the AST assays

98 is this the first time this customized card was used ? is this the same assay as used in the GEMs studies

114 “All HH stools positive for Shigella were not of the same species or serotype.” This sentence needs to ber explained. Not the same as what?

163 Explain “secondary education.” How many years of schooling is this?

189 I believe the ERC number should be included.

Reviewer #2: Minor comments/typographical:

Ln 37- I would favor replacing "vaccination" with " prevention" as the paper does not deal at all with vaccination and other prevention strategies are alluded to in the results (i.e reheating meals)

Ln 136- fill in the % value

Ln 186- Ethical clearance should go under methods

Fig 1- "CONSORT" should be capitalized

**Summary and General Comments**

Reviewer #1: This is an ambitious study to study shigella infections in households of index patients with shigellosis. It is an interesting study, but it seems the analysis could have gone into more depth in their analysis. For example:

1. When the study on household infections was being considered, did the authors have a hypothesis about the household infections? For example, did they want to determine if the infections were being spread between family members and if so, how? To answer this question, they would need to track the specific strains between household members. Perhaps alternatively, they just wanted to see if index cases identified households that have a high risk for shigellosis in general because of poor water/sanitation. If there was a hypothesis, what was the conclusion after doing the study?

2. What were the characteristics of the index cases? Did they all have dysentery or did some have watery diarrhea and were classified as an index case only after a positive culture was detected. Did the clinical symptoms change the rate of infections in household members?

3. What were the species and serotypes of the index cases, and how did these match the species / serotype of the infections in the household members. It would seem, there were many different strains of shigella in the household members, but they may not be related to the infection in the index cases.

4. What is the rate of shigella infection in the community? In this slum area of the city, it may be that shigella infections are common in the general population, especially infections that can be detected by PCR. Without a comparison of rates in a control group, it is difficult to know if the households had a higher rate than would be expected.

5. The analysis of households should be by household as well as individual. How many of the households had an infection. What was the distribution of the numbers of infections in the infected households. Did some households have many people infected and others had 0 or 1? It was not clear if the risk factors looked at households or only at individuals with infections.

6. I believe Table 1 can better describe the characteristics of the household members by age and sex.

7. Was there any seasonality among the index cases.

8. I believe the discussion section could be improved by first describing what the authors found in their study. I prefer to understand all the findings from this study first. Then, in later paragraphs compare their findings to other studies.

9. In terms of limitations, I would highlight the lack of control group. Ideally this could have been a case control study in which households of index patients with a non-diarrheal illness could be compared. Obviously, it is too late to do this now, but is there another way to estimate the rates in the community?

10. The antibiotic sensitivity patterns must have differed by species and serotype. By lumping together, I believe we do not have an appreciation of the true antibiotic sensitivity patterns.

Figure 2. I do not think a figure is needed. A table would be better.

Figure 3. Include the number of isolates tested

Figure 4. The figure does not have a label for the y axis. With so few pathogens identified with the Taq card, it would seem this does not deserve a figure. The data could be included in the text.

Reviewer #2: This manuscript describes a study assessing factors associated with Shigella carriage among household contacts of index children aged 6-35 months presenting with Shigella infection in Dhaka, Bangladesh. The cohort was relatively large, but still of limited size to detect differences in some factors. The methods were rigorously defined and support the conclusions drawn, which showed frequent asymptomatic carriage in household contacts and risk factors that included failure to reheat meals. I admit to being a little confused about how toilet type is shown to be significant in even the univariate analysis despite having only 1 infection in a total 2 individuals without flush toilets.

I think this paper does add to the general knowledge of household transmission of shigella and could be useful in developing strategies for screening or prevention.

PLOS authors have the option to publish the peer review history of their article (what does this mean? ). If published, this will include your full peer review and any attached files.

**Do you want your identity to be public for this peer review?** For information about this choice, including consent withdrawal, please see our Privacy Policy .

Reviewer #1: No

Reviewer #2: No

**Figure resubmission:**

**Reproducibility:**



---

## [Decision Letter · Decision Letter 1]

14 Oct 2025

Dear Dr. Qadri,

We are pleased to inform you that your manuscript 'Prevalence of Shigellosis among household contacts of index cases in the EFGH catchment area, Dhaka, Bangladesh' has been provisionally accepted for publication in PLOS Neglected Tropical Diseases.

Best regards,

Nicholas J Mantis

Academic Editor

Stuart Blacksell

Section Editor

Shaden Kamhawi

co-Editor-in-Chief

Paul Brindley

co-Editor-in-Chief

Reviewer's Responses to Questions

**Key Review Criteria Required for Acceptance?**

**Methods**

-Are the objectives of the study clearly articulated with a clear testable hypothesis stated?

-Is the study design appropriate to address the stated objectives?

-Is the population clearly described and appropriate for the hypothesis being tested?

-Is the sample size sufficient to ensure adequate power to address the hypothesis being tested?

-Were correct statistical analysis used to support conclusions?

-Are there concerns about ethical or regulatory requirements being met?

Reviewer #1: yes, the objectives are clear

**Results**

-Does the analysis presented match the analysis plan?

-Are the results clearly and completely presented?

-Are the figures (Tables, Images) of sufficient quality for clarity?

Reviewer #1: Yes

**Conclusions**

-Are the conclusions supported by the data presented?

-Are the limitations of analysis clearly described?

-Do the authors discuss how these data can be helpful to advance our understanding of the topic under study?

-Is public health relevance addressed?

Reviewer #1: Yes

**Editorial and Data Presentation Modifications?**

Reviewer #1: Figure legend is run-on with the Table 1

Table 2 has a wording problem in one field: "number of persons live per room"

I thought one weakness would be mentioned: no community control group to know the rate of shigella positives in the general population.

**Summary and General Comments**

Reviewer #1: This is much improved and the authors have addressed the concerns from my earlier review.

PLOS authors have the option to publish the peer review history of their article (what does this mean? ). If published, this will include your full peer review and any attached files.

**Do you want your identity to be public for this peer review?** For information about this choice, including consent withdrawal, please see our Privacy Policy .

Reviewer #1: No

---

## [Editor Report · Acceptance letter]

Dear Dr. Qadri,

We are delighted to inform you that your manuscript, "Prevalence of Shigellosis among household contacts of index cases in the EFGH catchment area, Dhaka, Bangladesh," has been formally accepted for publication in PLOS Neglected Tropical Diseases.

Best regards,

Shaden Kamhawi

co-Editor-in-Chief

Paul Brindley

co-Editor-in-Chief
